# Evaluation of ^68^Ga-Radiolabeled Peptides for HER2 PET Imaging

**DOI:** 10.3390/diagnostics12112710

**Published:** 2022-11-05

**Authors:** Maxwell Ducharme, Hailey A. Houson, Solana R. Fernandez, Suzanne E. Lapi

**Affiliations:** Department of Radiology, University of Alabama at Birmingham, Birmingham, AL 35294, USA

**Keywords:** PET, HER2, peptides, imaging, breast cancer

## Abstract

One in eight women will be diagnosed with breast cancer in their lifetime and approximately 25% of those cases will be HER2-positive. Current methods for diagnosing HER2-positive breast cancer involve using IHC and FISH from suspected cancer biopsies to quantify HER2 expression. HER2 PET imaging could potentially increase accuracy and improve the diagnosis of lesions that are not available for biopsies. Using two previously discovered HER2-targeting peptides, we modified each peptide with the chelator DOTA and a PEG_2_ linker resulting in DOTA-PEG_2_-GSGKCCYSL (P5) and DOTA-PEG_2_-DTFPYLGWWNPNEYRY (P6). Each peptide was labeled with ^68^Ga and was evaluated for HER2 binding using in vitro cell studies and in vivo tumor xenograft models. Both [^68^Ga]P5 and [^68^Ga]P6 showed significant binding to HER2-positive BT474 cells versus HER2-negative MDA-MB-231 cells ([^68^Ga]P5; 0.68 ± 0.20 versus 0.47 ± 0.05 *p* < 0.05 and [^68^Ga]P6; 0.55 ± 0.21 versus 0.34 ± 0.12 *p* < 0.01). [^68^Ga]P5 showed a higher percent injected dose per gram (%ID/g) binding to HER2-positive tumors two hours post-injection compared to HER2-negative tumors (0.24 ± 0.04 versus 0.12 ± 0.06; *p* < 0.05), while the [^68^Ga]P6 peptide showed significant binding (0.98 ± 0.22 versus 0.51 ± 0.08; *p* < 0.05) one hour post-injection. These results lay the groundwork for the use of peptides to image HER2-positive breast cancer.

## 1. Introduction

One in eight women will be diagnosed with breast cancer in their lifetime and 20–30% of these will be Human Epidermal Growth Factor 2 (HER2)-positive [1]. HER2-positive breast cancer shows an increased expression of HER2 protein, which is associated with aggressive tumor growth and high metastasis rate [1]. The increase in HER2 protein expression in cancer can be between 40 to 100 times greater than the natural expression of HER2 [2,3]. HER2 is one of a family of four membrane tyrosine receptor kinases (HER1-HER4); however, it is the only receptor in the family that lacks any natural ligands [4,5]. The HER family initiates a cellular response by ligand–receptor binding which induces either hetero- or homodimerization between other members of the family or the same receptor type as itself [1]. However, since there is no natural ligand for HER2, when overexpressed, the ability to homodimerize without a ligand present is increased, leading to the uncontrolled activation of cellular pathways, including the MAPK and PI3K pathways, leading to proliferation and growth [6]. HER2 is also the preferred dimerization partner for the other HER receptors [1,2].

Current methods to diagnose HER2-positive breast cancer involve biopsied tissue. Immunohistochemistry (IHC) is used to stain and semi-quantitatively measure HER2 protein expression with a 0-3+ scale indicating little to no expression (0) to highly overexpressing (3+). Fluorescence in situ hybridization (FISH), which utilizes silver staining of the HER2 gene, is completed independently of IHC, however, and offers a second complementary technique to determine HER2 expression [7]. While no HER2 expression and highly over-expressive HER2 status tend to yield relatively clear results using these methods, biopsies of tumor tissue that have moderate expression can create challenges in accurately diagnosing HER2 quantity. Heterogeneity in singular tumors, along with variable HER2 status between metastatic lesions, can also limit the understanding of the full picture of a patient’s disease when looking at a single biopsy sample [7,8]. Moreover, taking multiple biopsies may not be possible depending on their location and the increased discomfort to the patient. A strategy that does not require biopsies and has the potential to view all metastatic sites is HER2 imaging with Positron Emission Tomography (PET). PET utilizes positron-emitting isotopes combined with targeting molecules that can image important biomarkers of cancer and other diseases. The use of PET imaging can aid in diagnosing cancer patients with HER2-positive lesions, leading to increased diagnostic accuracy and optimal treatments.

HER2 PET imaging is currently being explored in phase II clinical trials involving two different antibodies, trastuzumab (Herceptin^®^) and pertuzumab (Perjeta^®^) [9,10,11]. These antibodies are used as first-line treatments for HER2-positive breast cancer but can be used as imaging agents when paired with a positron-emitting radioisotope, in this current case, ^89^Zr. ^89^Zr decays 23% by positron emission, 77% by electron capture, and has a half-life of 3.27 days [12]. Though antibodies have a high affinity for their targets, some limitations are associated with antibody-based imaging. One major hurdle involves the long time required between injection and imaging, typically 3–7 days [11]. This is due to the long blood circulation time of antibodies, which require days to allow for the background radioactivity within the blood and non-target organs to diminish. This long imaging time requires longer-lived radioisotopes, such as ^89^Zr, with half-lives of days to pair with the antibodies, increasing the dose to the patient. A potential alternative to antibody-based PET imaging agents to improve upon the time required and potential dose to a patient is the use of peptide-based imaging agents [13].

Peptide imaging agents allow for quicker injection to imaging times along with a potential decrease in dose to the patient by using a shorter-lived radioisotope. Radiolabeled peptides are currently used in the clinic for the diagnosis of several cancer types. A widely used example is [^68^Ga]DOTATATE (NETSPOT^®^) which was given FDA approval in 2016 and is used for imaging somatostatin receptor type 2 on neuroendocrine cancer with patient imaging 40 to 90 min after intravenous administration [14,15,16]. Several HER2-targeting peptides have been reported which could have the potential to be used for PET imaging agents. Originally discovered by Karasseva et al. using a six amino-acid bacteriophage display library, the KCCYSL peptide motif has been evaluated for its binding to HER2 and showed an equilibrium dissociation constant of 30 µM, indicating its potential use for PET imaging [17,18]. Since then, the motif has been incorporated into various peptides to yield agents with improved affinity, stability, and overall targeting to HER2 for single-photon emission computed tomography (SPECT) imaging [19,20,21]. In another example, using a one-bead-one-compound (OBOC) method, Geng et al. found 72 peptides that could bind to HER2. The peptide DTFPYLGWWNPNEYRY from their library showed the highest affinity (18.6 nM) for HER2 and illustrated HER2-positive tumor targeting in vivo using fluorescence imaging [22].

Considering the advantages associated with peptide-based imaging probes and the availability of multiple peptides that target HER2, our goal was to develop PET imaging radiopeptides that can rapidly image HER2-positive tumor lesions. We hypothesized that our modified HER2-targeting peptides would have high radiolabeling efficiency using the radioisotope ^68^Ga, selectively bind to HER2 in vitro, and show significantly higher binding in mice bearing HER2-positive tumors versus HER2-negative tumors.

## 2. Materials and Methods

### 2.1. Chemicals and Reagents

Unless otherwise stated, all chemicals were purchased from Thermo Fisher Scientific (Waltham, MA, USA). Sodium acetate, hydrochloric acid (HCl), trifluoroacetic acid (TFA), and sodium hydroxide (NaOH) were purchased from Millipore Sigma (Burlington, MA, USA). Peptides were synthesized and purchased from CPC Scientific Co., (San Jose, CA, USA). Phosphoramidon was purchased from Selleck Chemicals LLC (Houston, TX, USA). Thermolysin enzyme was purchased from R&D Systems Inc. (Minneapolis, MN, USA).

### 2.2. Cell Culture

MDA-MB-231 (HER2−) and BT474 (HER2+) cells were purchased from American Type Culture Collection (Manassas, VA, USA). Both cell lines were grown in Gibbco’s high glucose Dulbecco’ Modified Eagle’s Medium (DMEM) supplemented with 10% fetal bovine serum (FBS) and 80 μM gentamicin. The BT474 cell line media also had the addition of 1.8 mM of insulin. All cells were maintained and grown in humidified incubators at 37 °C with 5% CO_2_. DMEM media, FBS, and gentamicin were purchased from Thermo Fisher, while the insulin was purchased from Millipore Sigma.

### 2.3. Radiolabeling

The P5 (GSGKCCYSL) and P6 (DTFPYLGWWNPNEYRY) peptides were synthesized with the chelator 2,2′,2″,2‴-(1,4,7,10-Tetraazacyclododecane-1,4,7,10-tetrayl) tetraacetic acid (DOTA) and a polyethylene glycol (PEG_2_) linker between the chelator and the amino acid sequences. Each peptide was dissolved in DMSO to a concentration of 2 μg/μL. Peptides were radiolabeled with ^68^Ga with a procedure adapted from the literature methods [23,24,25,26]. ^68^Ga was eluted from an Eckert and Ziegler ^68^Ge/^68^Ga generator using 0.1 M HCl and was concentrated on an Agilent Bond Elut SCX cartridge. ^68^Ga was removed from the cartridge using 200 μL of a 5 M NaCl/0.1 M HCl solution at a concentration of 1.11–2.22 MBq/μL. An amount of 30 μg of peptide (18.6 μmol P5; 10.6 μmol P6), 37 MBq of ^68^Ga (~22 µL), and an equal volume of 1M sodium acetate pH 4.5 were combined in a microcentrifuge tube. The solution was adjusted to a pH of 4.5 with either 1M HCl or NaOH and incubated in a thermomixer at 95 °C for 12–15 min while shaking at 1200 rpm. Samples were analyzed by an Agilent Technologies HPLC Infinity 1260 model (Santa Clara, CA, USA) equipped with a multi-wavelength diode array detector and Flow-RAM Galbi NaI (Lab Logic, Tampa, FL, USA) using a Zorbax Stablebond 300 C18, 4.6 × 150 mm, 3.5 μm HPLC column. The HPLC gradient was 20 min, starting with 100% water with 0.1% trifluoroacetic acid (TFA). After two minutes of 100% water, the gradient increased from 0%–100% acetonitrile (ACN) with 0.1% TFA over 10 min. The gradient ran with 100% ACN for 3 min before returning to 100% water over 2 min. The final 3 min ran with 100% water to prepare for the next sample.

### 2.4. Degradation Studies

In a microcentrifuge tube, 15 μg (~35 μL) of the radiolabeled peptide was combined with 35 μL of enzyme buffer (25 mM Tris, 150 mM NaCl, and 10 mM CaCl_2_) and 14 pmol of thermolysin, a common metalloendopeptidase that is excreted by the gram-positive bacteria Bacillus thermoproteolyticus [27]. Samples were incubated at 37 °C while shaking at 800 rpm. Every 30 min, 10 μL samples were taken and analyzed via HPLC. In a separate vial, 2 nmol of phosphoramidon (PA), a non-specific protease inhibitor, was added to the same mixture under the conditions described above to evaluate the ability to slow peptide degradation [28]. Samples were analyzed via HPLC as described previously with an extended gradient. The extended gradient started with 100% water for 2 min, followed by a 20-min gradient increase to 100% ACN. The 100% ACN was run for 3 min before returning to 100% water over 2 min. The final 3 min ran with 100% water to prepare for the next sample. Serum studies combined 15 μg (~35 μL) of [^68^Ga]P5 or [^68^Ga]P6, 300 μL of mouse or human serum, and 1 μM of phosphoramidon (if appropriate group) shaking at 800 rpm at 37 °C. Every h, 30 μL of serum with radiolabeled peptide was added to 30 μL of ACN and was vortexed and spun down to analyze 30 μL of supernatent on the HPLC using the 20-min gradient as described previously. ROIs were drawn to compare the standard radiolabeled peak of each peptide before serum addition.

### 2.5. Cell Binding

Each cell line was plated on 6-well plates at a concentration of 1 × 10^6^ cells per well and allowed to adhere overnight. For P5 cell studies, the peptide was radiolabeled as described above and diluted in complete media for a final concentration of 300 nM. The P6 cell studies used 250 nM concentration of radiolabeled peptide and were diluted in phosphate buffer saline (PBS). Media or PBS containing the radiopeptide was supplemented with 1 μM of phosphoramidon to slow degradation. Each well was incubated with 1 mL of radiopeptide solution for either 15 or 30 min at 37 °C for P5 and P6, respectively. After incubation, the radiopeptide solution was aspirated and each well was washed with 1 mL of ice-cold PBS in triplicate. To remove cells, 500 μL of 0.5 M NaOH was added and allowed to incubate for 5 min. Single-well components were transferred in individual microcentrifuge tubes along with a 500 μL PBS wash and placed on a Hidex AMG automated gamma counter to determine the amount of radioactivity in each sample. Values were normalized to the overall protein amount determined by a BCA assay.

### 2.6. Mouse Biodistribution and Imaging Studies

Female athymic nude mice were purchased from Charles Rivers Laboratories (Wilmington, MA, USA) or Jackson Laboratories (Bar Harbor, ME, USA). After one week of acclimation, mice designated for BT474 tumors were implanted with a 20 mg homemade pellet containing 0.72 mg of β-estradiol with cholesterol subcutaneously in the right shoulder to aid in tumor growth. One week after the pellet implantation, mice were injected with 10 million BT474 cells in a 1:1 ratio of matrigel and PBS subcutaneously in the left shoulder. Mice that did not receive the pellet were implanted with 10 million MDA-MB-231 cells. On average, palpable tumors (approximately 5 × 5 × 5 mm) developed 4 to 6 weeks post-cell implantation. On the day of the study, approximately 5 μg (3.7 MBq) of either [^68^Ga]P5 or [^68^Ga]P6 with 300 μg of phosphoramidon was diluted with sterile water to a calculated osmolality of approximately 300 mOsm. If needed, 0.9% sterile saline was added to achieve 100 μL injections. Mice were anesthetized with 2.5% isoflurane in oxygen and were injected via a tail vein with the radiolabeled peptide and PA. Mice were allowed to roam in their cages for 1 or 2 h before being anesthetized for imaging. Imaging was conducted on a Sofie GNEXT PET/CT. In total, 20 min of PET data were acquired for mice 1 h post-injection while 30 min of data were acquired for mice 2 h post-injection. Each PET imaging session was followed by a 5-min CT at 80 kVp. Two different cohorts of mice, each bearing MDA-MB-231 or BT474 tumors, were also injected with ~1.85 MBq of [^68^Ga]P6 and PET images were collected dynamically for 90 min with frame reconstruction at 10 × 60 s, 4 × 300 s, and 6 × 600 s followed by a 5-min CT. After imaging, mice were sacrificed and organs were dissected, weighed, and counted for radioactivity on a Hidex AMG automated gamma counter. Uptake of radioactivity was calculated as percent injected dose per gram of tissue (%ID/g).

### 2.7. Image Analysis

PET images were reconstructed via 3D-OSEM (Ordered Subset Expectation Maximization) algorithm (24 subsets and 3 iterations), with random, attenuation, and decay correction, and CT was reconstructed with Modified Feldkamp Algorithm and analyzed using VivoQuant (Invicro) software. After the images were reconstructed, standard uptake values (SUVs) were determined by hand-drawing regions of interest (ROIs) from both the HER2-positive and HER2-negative tumors and an adjacent muscle within the same mouse using CT anatomical guidelines. Radioactivity in each ROI was calculated as SUV (Equation (1)).
(1)$$SUV=\frac{Concentration\;of\;Radioactivity\;in\;Tumor\;\left(\frac{MBq}{mL}\right)}{Injected\;Radioactivity\;\left(MBq\right)\;/\;mouse\;weight\;\left(g\right)\;}$$

### 2.8. Statistical Analysis

Quantitative analysis was expressed as mean ± SD. Comparisons were made using Prism 8 software running Student’s *t*-test. *p* values of less than 0.05 were considered significant.

## 3. Results

### 3.1. Radiolabeling of DOTA Conjugated Peptides with ^68^Ga

With the addition of DOTA and the PEG_2_ linker, the peptides had final sequences of DOTA-PEG_2_-GSGKCCYSL (P5) and DOTA-PEG_2_-DTFPYLGWWNPNEYRY (P6) (Appendix A). Our labeling method resulted in high radiolabeling percentages (>95%), leading to average molar activities of 2.8 MBq/nmol and 1.7 MBq/nmol for P5 and P6, respectively (Figure 1). This protocol was followed for all in vitro and in vivo studies to confirm >95% labeling via HPLC before moving forward.

### 3.2. Degrading [^68^Ga]Peptides Using Thermolysin

Radiolabeled peptides were subjected to proteolytic degradation from thermolysin as described above to determine behavior in vivo if unprotected. [^68^Ga]P5 and [^68^Ga]P6 showed significant degradation after 30, 60, and 90 min after thermolysin exposure. The ^68^Ga radiochromatogram shows the degradation products that are detected before the original intact radiopeptide peak (Figure 2). [^68^Ga]P5 had an average of 87.7% degradation, while [^68^Ga]P6 had an average of 47.5% degradation after 90 min. Our results show that both [^68^Ga]P5 and [^68^Ga]P6 are susceptible to protease degradation.

### 3.3. Delaying Degradation Using Phosphoramidon

Due to the degradation of these peptides from one protease, it is expected that the peptides would potentially be degraded in in vivo studies. Thus, phosphoramidon (PA) was added to slow the degradation of the peptide. Phosphoramidon is a non-specific protease inhibitor that has been shown to help slow the degradation of peptides composed of natural amino acids in vivo [28]. The addition of phosphoramidon slowed the degradation of both [^68^Ga]P5 and [^68^Ga]P6 in the presence of thermolysin for up to 90 min.

There was a significant difference in the amount of intact peptide starting as early as 30 min after incubation (*p* < 0.0005 and *p* < 0.005 for [^68^Ga]P5 and [^68^Ga]P6, respectively) which was maintained at 60 and 90 min compared to the thermolysin-only samples (Figure 3). The degradation of [^68^Ga]P5 showed multiple products before the initial retention time of the control [^68^Ga]P5 peak, while [^68^Ga]P6 showed only two major degradation products.

Serum studies with both [^68^Ga]P5 and [^68^Ga]P6 with and without phosphoramidon were conducted for up to 3 h in both mouse and human serum. Overall, the [^68^Ga]P5 peptide showed slowed degradation with the use of phosphoramidon in both mouse and human serum with over 80% of the peptide intact up to 3 h, while [^68^Ga]P6 had no slowed degradation with the use of phosphoramidon in either mouse or human serum. However, appoximately 20% of [^68^Ga]P6 was intact after 1 h in mouse serum whereas no signs of degradation were seen in human serum (Appendix A). Phosphoramidon was able to slow the degradation of [^68^Ga]P5, indicating its potential in in vivo application as first shown by Nock et al. [28].

### 3.4. [^68^Ga]P5 and [^68^Ga]P6 Show Binding to HER2 In Vitro

[^68^Ga]P5 and [^68^Ga]P6 peptides were labeled with 37 MBq of ^68^Ga with radiolabeling efficiency greater than or equal to 95% via HPLC. [^68^Ga]P5 had a significantly higher mean uptake of 0.68 ± 0.20 in BT474 cells compared to 0.47 ± 0.05 percent activity per mg protein in MDA-MB-231 cells, respectively (*p* < 0.05) (Figure 4a).

[^68^Ga]P6 also showed higher and significant binding to BT474 cells versus MDA-MB-231 cells (Figure 4b) (0.55 ± 0.21 versus 0.34 ± 0.12; *p* < 0.01). Each peptide illustrated higher binding to the HER2-positive cells compared to the HER2-negative cells.

### 3.5. Mice Tumor Xenograft Models Show Increased Peptide Binding to Tumors Expressing HER2

[^68^Ga]P5 and [^68^Ga]P6 with phosphoramidon were injected via a tail vein in mice bearing either HER2-negative or HER2-positive tumors. [^68^Ga]P5 was injected and mice were imaged 2 h post-injection followed immediately by the biodistribution study.

[^68^Ga]P5 had significantly higher %ID/g in HER2-positive tumors compared to HER2-negative tumors (0.24 ± 0.04 versus 0.12 ± 0.06; *p* < 0.05; Figure 5). [^68^Ga]P5 was also evaluated at 1 h post-injection and showed a lower tumor-to-blood ratio in HER2-positive mice compared to the 2 h timepoint (0.37 versus 1.01). At 1 h post-injection, no difference was observed between the HER2-negative and HER2-positive tumor-to-blood ratios (0.39 and 0.37, respectively), whereas at 2 h post-injection, HER2-positive mice showed close to a four times higher ratio compared to the HER2-negative mice at 1.01 and 0.27, respectively (Appendix A).

[^68^Ga]P6 was injected into athymic nude mice bearing HER2-positive tumors (BT474 cells) and HER2-negative (MDA-MB-231 cells) and imaged 1 h post-injection. HER2-positive tumors had an uptake of 0.98 ± 0.22, while HER2-negative tumors had a significantly less uptake of 0.51 ± 0.08; *p* < 0.05 (Figure 6). Regardless of tumor group, all animals showed the highest percent injected dose per gram (%ID/g) in the kidneys, which was expected as the peptide is excreted through the kidneys, with low uptake present in other vital organs. Compared to the 2 h post-injection biodistribution, higher tumor-to-blood ratios were obtained for both HER2-negative (0.74 versus 0.36) and HER2-positive (0.89 versus 0.55) at the 1 h timepoint (Appendix A).

ROI analysis of [^68^Ga]P6 showed a positive but non-significant trend in tumor-to-muscle ratio after 1 h for the HER2-positive mice compared to the HER2-negative mice (1.37 ± 0.30 versus 1.20 ± 0.18; *p* = 0.36; *n* = 4). Images of the selected frame showed differences in peptide binding in the tumors (Figure 7). SUV analysis illustrated that the HER2-positive tumors had an average SUV of 0.72 ± 0.16 compared to the HER2-negative tumors with an SUV of 0.61 ± 0.19; however, the difference is not significant. Sagittal and coronal slices in relation to the transverse slice are included in Appendix A.

## 4. Discussion

The use of PET imaging has shown potential within a research setting to be a valuable asset to non-invasively determine if breast cancer is HER2-positive. The aggressive nature of HER2-positive breast cancer, as well as its poor prognosis, makes its early detection and rapid treatment vital for patient outcomes. A variety of probes have been evaluated for PET imaging pre-clinically and in clinical trials for the detection of HER2-positive lesions. The radiolabeled antibodies [^89^Zr]trastuzumab and [^89^Zr]pertuzumab are currently in clinical trials and have an extremely high affinity for HER2 [9,10]. However, antibodies have long blood circulatory times which require days to allow for clearance from the blood before PET acquisition can take place. Longer-lived isotopes like ^89^Zr are also required to match the days required to acquire an image, leading to an increased dose to a patient [13,29]. Peptides may offer solutions to these drawbacks with the potential for quicker tissue penetration within hours, allowing for a shorter-lived radioisotope to be used [25].

We chose the KCCYSL (P5) peptide motif as it has been evaluated pre-clinically with SPECT imaging but had yet to be evaluated for PET imaging using ^68^Ga [17,18,19,20]. The development as a PET agent from a SPECT agent would potentially increase the spatial resolution, which is important for the imaging of small legions including metastasis. The DTFPYLGWWNPNEYRY (P6) peptide was recently discovered by Geng et al. and had a reported Kd of 18.6 nM [22]. The peptide had yet to be evaluated for PET imaging so, with the favorable K_d_, we hypothesized that the peptide would have potential as a PET imaging agent.

Radiolabeling our DOTA-conjugated P5 and P6 peptides with ^68^Ga resulted in efficient radiolabeling and high molar activities of 2.8 MBq/nmol and 1.7 MBq/nmol for [^68^Ga]P5 and [^68^Ga]P6, respectively. These peptides were radiolabeled with ^68^Ga, which is a favorable radioisotope for PET imaging since it has a high positron branching ratio of 89% and can be produced without a cyclotron using a ^68^Ge/^68^Ga generator [30]. The rapid radiolabeling procedure (<20 min) is optimal to minimize the decay of activity since ^68^Ga has a half-life of only 68 min [23,30]. The short half-life of ^68^Ga also decreases the dose a patient will receive compared to ^89^Zr.

One drawback for peptides is their susceptibility to degradation from proteases present within organisms. Using the non-specific protease inhibitor phosphoramidon, as investigated by Nock et al., we showed that we could slow the degradation of our [^68^Ga]P5 and [^68^Ga]P6 peptides by up to 90 min in the presence of a common protease. Phosphoramidon can be used as an aid to protect peptides from quick degradation when evaluating our peptides in vivo. Suda et al. reported that up to a 1 g/kg intravenous injection showed low toxicity and no death in mice [31]. We are using phosphoramidon as a tool to slow the degradation of our peptides to evaluate them thoroughly in vivo. Studies involving human use would need to be conducted to confirm no toxicity to humans. Our serum studies showed a similar trend using mouse and human serum. [^68^Ga]P5 showed >80% of the intital intact peptide up to 3 h in both mouse and human serum with the addition of phosphoramidon (Figure 4a,b). However, [^68^Ga]P6 showed to only be 20% intact after 1 h with phosphoramidon in mouse serum, while showing no signs of degradation in human serum up to 3 h (Appendix A) The rapid degradation of [^68^Ga]P6 in mouse serum shows the need for peptide optimization in the future to increase the in vivo stability.

To confirm selective binding to HER2, cells expressing HER2 (BT474) and HER2-negative control cells (MDA-MB-231) were analyzed after incubation with [^68^Ga]P5 and [^68^Ga]P6. While overall binding was modest, both peptides showed a significantly higher percentage of activity per mg of protein in HER2-positive cells versus HER2-negative cells. However, due to this modest binding, cell-binding studies with excess unlabeled peptide showed no significant difference between blocking and non-blocking groups (Appendix A). These studies were conducted with the aid of phosphoramidon due to proteases that are present on the surface of cells that could degrade peptides and prevent their interaction with HER2 [32].

[^68^Ga]P5 and [^68^Ga]P6 were injected into female athymic nude mice to determine tumor uptake and off-target binding. Mice bearing BT474 tumor cells were implanted with a home-made pellet to secrete estrogen to enhance tumor growth. The pellet did not show any uptake of either peptide and was not expected to alter any other biological processes within the mice. As reported by Nock et al., using phosphoramidon as a co-injection with [^68^Ga]P5 and [^68^Ga]P6 was expected to increase the in vivo stability of the peptides, extending their circulatory time and increasing their chance to bind to HER2 [28]. [^68^Ga]P5 showed the highest contrast between HER2-positive and HER2-negative tumors 2 h post-injection (0.24 ± 0.04% ID/g versus 0.12 ± 0.06% ID/g; *p* < 0.05). The high uptake in the kidney was expected due to the excretion of peptides through the kidneys and was confirmed with the high accumulation of radioactivity within the bladder of each mouse. The blood showed a low amount of peptide present (<0.5 %ID/g), indicating that there was little peptide still in blood circulation. [^68^Ga]P6 showed the highest contrast between HER2 status 1 h post-injection (0.98 ± 0.22 %ID/g versus 0.51 ± 0.08 %ID/g; *p* < 0.05). However, SUV analysis of the PET images of these mice showed no statistical difference between HER2-positive and negative tumors, potentially indicating the lower sensitivity of the PET imaging as compared to the biodistribution data. This modest uptake is likely partially the result of some degradation of [^68^Ga]P6 in vivo as illustrated by the serum stability studies. [^68^Ga]P5 was also evaluated at 2 h post-injection without the use of phosphoramidon in the HER2-positive tumor mice. With the co-injection of phosphoramidon, there was an increase of tumor-to-blood from 0.77 versus 1.0, supporting the use of phosphoramidon with peptides susceptible to protease degradation (Appendix A). Though both peptides have their advantages, the P6 peptide likely has more potential for clinical use. The P6 peptide showed higher tumor uptake over the P5 peptide 1 h earlier with no signs of degradation in human serum, indicating that a translation to human use without the use of phosphoramidon may be achievable. Despite similar binding in our cell-binding experiments, the P6 peptide has a much lower reported K_d_ at 18.6 nM, which is ideal for radiotracer development. Though these peptides showed higher binding to HER2-positive tumors compared to HER2-negative tumors in biodistribution studies, future work is required to optimize these agents for PET imaging.

The overall higher %ID/g in the [^68^Ga]P6 mice compared to the [^68^Ga]P5 mice is due to two main factors. The biodistribution was done 1 h earlier, leading to more [^68^Ga]P6 in blood circulation and, as [^68^Ga]P6 is more lipophilic, it is more likely to stick non-specifically (and transiently) to blood proteins. Both peptides showed significantly higher binding to HER2-expressing tumors compared to the HER2-negative tumors; however, the difference in the overall binding and the difference in binding between the tumors was modest. While this is the first report of a ^68^Ga-labeled version of the P5 peptide, comparing our P5 peptide with other peptides with the KCCYSL motif, uptake in HER2-positive tumors varied. Uptake with MDA-MB-435 tumors shown by Kumar et al. and Larimer et al., and SKOV3 cells from Deutscher et al., had a range of 0.34 ± 0.03 %ID to 0.66 ± 0.11% ID/g 2 h post-injection [18,19,20]. We show an uptake of 0.24 ± 0.04% ID/g, which was slightly lower than the reported range. Our P6 peptide, first discovered by Geng et al., had yet to be evaluated for imaging for either SPECT or PET imaging [22]. Another HER2-targeting peptide (LTVSPWY) has been evaluated for both SPECT and PET imaging. Sabahnoo et al. reported two different peptides, CGGG-LTVSPWY and CSSS-LTVSPWY, had the ability to target HER2-positive SKOV-3-derived tumors [33]. At 1 h post-injection, the CGGG-LTVSPWY had a tumor %ID/g of 3.84 ± 2.53, while the CSSS-LTVSPWY peptide had an uptake of 4.95 ± 4.84 with the radioisotope ^99m^Tc. While tumor uptake was higher compared to our peptides, intestine uptake was greater than 20% ID/g, along with kidney values greater than 70% ID/g, and blood values greater than 3% ID/g. ^68^Ga-DOTA-(Ser)_3_-LTVSPWY was also evaluated using the SKOV-3 tumor model. Tumor %ID/g at 1 h post-injection was 0.56 ± 0.26 and 0.26 ± 0.02 at 2 h [34]. These values are similar to both of our peptides at each respective timepoint. However, with all of these evaluated peptides, only HER2-positive models were used, while we were able to show significant differences between HER2-positive and HER2-negative models in vivo. Recently, HER2-targeting camelid single-domain antibody fragments (sdAb, nanobodies) have been in development. These nanobodies are a tenth of the size of an antibody (~15 kDa), giving nanobodies greater tissue penetration and rapid blood clearance for short injection-to-image time within hours [35,36,37]. While high tumor uptake has been reported after 1 h (4–18% ID/g) using both ^68^Ga and ^18^F labeled nanobodies, extremely high kidney retention points to the requirement of more research to improve nanobodies for future use.

These peptides could be further optimized by the addition of pseudopeptide bonds or unnatural amino acids. These additions make it more difficult for proteases to recognize peptide bonds and specific amino acid sites to degrade. However, receptor affinity would need to be re-evaluated after alterations to make sure essential binding sites were not altered. Additionally, peptides with higher binding affinities could be developed. With significant uptake within hours and the use of a shorter-lived isotope, the [^68^Ga]P5 and [^68^Ga]P6 peptides have the potential for continued development for HER2 PET imaging.

## 5. Conclusions

[^68^Ga]P5 and [^68^Ga]P6 have exhibited potential for their use as HER2-targeting agents for PET imaging. Radiolabeling of both peptides yielded high specific activity with rapid radiolabeling times. Specific HER2 binding was exhibited using both in vitro cell-binding studies and in vivo tumor-binding studies. Further work is needed to increase the stability of the peptides and limit off-target binding. The ability to have significant binding differences within 1 or 2 h after injection shows that HER2-targeting peptides should continue to be evaluated for future use in PET imaging.

## Figures and Tables

**Figure 1 diagnostics-12-02710-f001:**
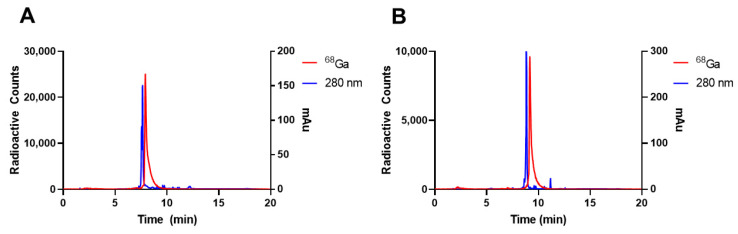
Radioactivity and 280 nm UV HPLC chromatograms of (**A**) P5 (DOTA-PEG_2_-GSGKCCYSL) and (**B**) P6 (DOTA-PEG_2_-DTFPYLGWWNPNEYRY). [^68^Ga]P5 was synthesized with a molar activity of 2.8 MBq/nmol with an average UV retention time of 7:39 min. [^68^Ga]P6 was synthesized with a molar activity of 1.7 MBq/nmol with an average UV retention time of 8:51 min.

**Figure 2 diagnostics-12-02710-f002:**
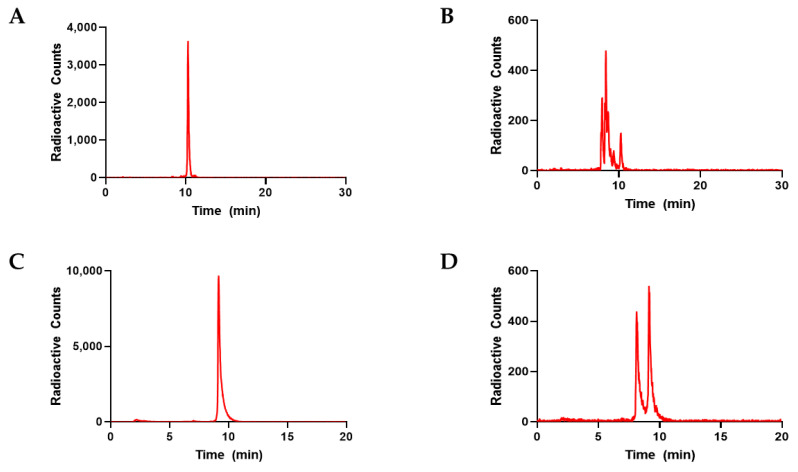
HPLC ^68^Ga chromatograms showing (**A**) [^68^Ga]P5 peptide at time zero before incubation with thermolysin. (**B**) [^68^Ga]P5 degradation after 90 min incubated with thermolysin. (**C**) [^68^Ga]P6 at time zero. (**D**) [^68^Ga]P6 degradation after 90 min incubated with thermolysin.

**Figure 3 diagnostics-12-02710-f003:**
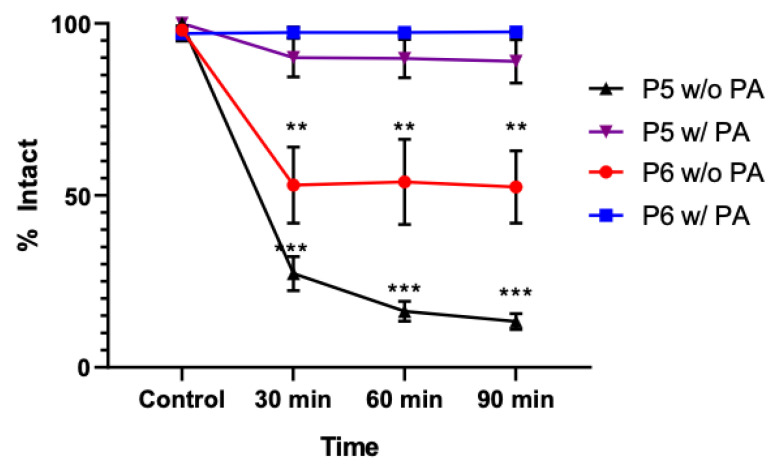
Line graph showing the degradation of both [^68^Ga]P5 and [^68^Ga]P6 when exposed to thermolysin with or without phosphoramidon. There was significant degradation up to 90 min for both [^68^Ga]P5 and [^68^Ga]P6 without the aid of phosphoramidon. ** *p* < 0.005 and *** *p* < 0.0005.

**Figure 4 diagnostics-12-02710-f004:**
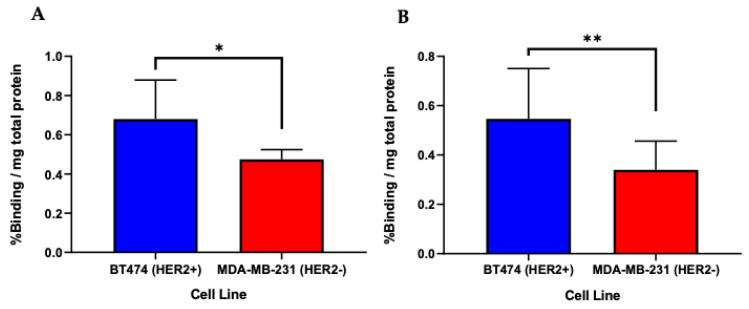
Bar graphs representing [^68^Ga]P5 and [^68^Ga]P6 binding to BT474 (HER2+) and MDA-MB-231 (HER2−) cells. (**A**) [^68^Ga]P5 had significantly higher % binding per mg of protein to BT474 cells (0.68 ± 0.20) versus MDA-MB-231 cells (0.47 ± 0.05) after 15 min of incubation. (*n* = 6) (**B**) [^68^Ga]P6 also had significantly higher binding to BT474 cells (0.55 ± 0.21) versus MDA-MB-231 cells (0.34 ± 0.12) after 30 min. (*n* = 12) * *p* < 0.05 and ** *p* < 0.01.

**Figure 5 diagnostics-12-02710-f005:**
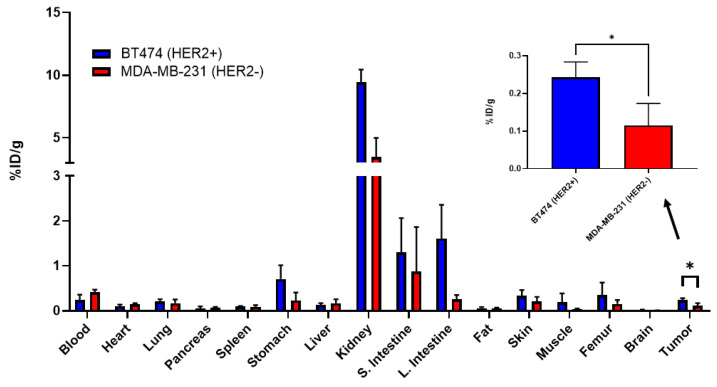
Biodistribution of [^68^Ga]P5 two hours post-injection in athymic nude mice bearing HER2-negative (MDA-MB-231 cells) and HER2-positive tumors (BT474 cells). Each organ was placed on the HIDEX AMG to determine the amount of [^68^Ga]P5 uptake. HER2-positive tumors showed significantly higher binding than the HER2-negative tumors (*n* = 4). * *p* < 0.05.

**Figure 6 diagnostics-12-02710-f006:**
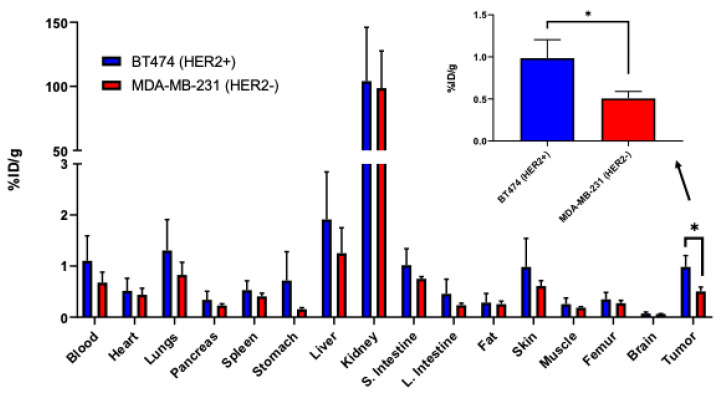
One hour post-injection biodistribution for [^68^Ga]P6 in athymic nude mice. [^68^Ga]P6 showed significantly higher binding to HER2-positive tumors (*n* = 4) versus HER2-negative tumors (*n* = 3). * *p* < 0.05.

**Figure 7 diagnostics-12-02710-f007:**
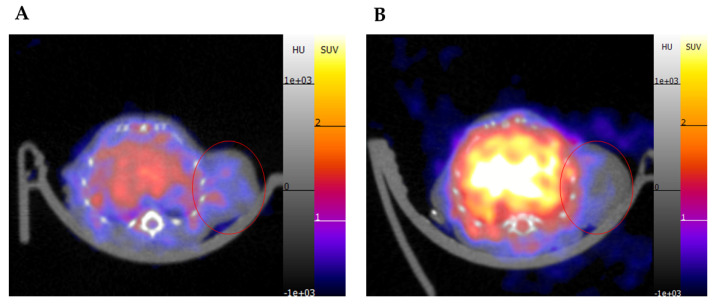
PET images of the 50-60 min frame with a SUV scale of 0.2-3. (**A**) [^68^Ga]P6 showed stronger binding to the HER2-positive tumor with an SUV mean value of 0.72 ± 0.16, while (**B**) less binding was seen in the HER2-negative tumor shown by the lower SUV mean value of 0.61 ± 0.19.

## Data Availability

The data presented in this study are available on request from the corresponding author.

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
