# Peer review of "Evaluation of 68Ga-Radiolabeled Peptides for HER2 PET Imaging"

_diagnostics, 2022, doi:10.3390/diagnostics12112710_

Round 1

Reviewer 1 Report (Previous Reviewer 2)

This manuscript reported the development and evaluation of 68Ga-radiolabeled peptides. This study was well performed and improved, but it still remains unclear whether these probes really work for visualizing HER2 high-expressing tumors in vivo. Thus, it should be reconsidered and revised in accordance with my opinion written below.

1. In the in vivo biodistribution study, the accumulation levels in BT474 tumor were higher than those in MDA-MB-231 tumor. Still, the levels in the other organs were also different depending on the mice types. Thus, it still remains hard to confirm if the different accumulation levels in tumor tissues depended on the HER2 expression levels of the tumors and not be affected by the pretreatment of beta-estradiol. Therefore, the biodistribution study should be performed with a mice model xenografting BT474 and MDA-MB-231 to avoid estimating the normal organs and the effect of the pretreatment.

2. Regarding PET imaging study, this manuscript showed only axial images, but MIP or coronal images would be more helpful to understand the distribution of those probes in the whole body.

3. According to the results of the PET imaging (figure 7), the distribution of [68Ga]P6 seems to be heterogeneous in the tumor tissues. Thus, it would be helpful to confirm the distribution of the probe was dependent on the HER2 expression by comparing autoradiography and HER2 immunohistochemistry with serial frozen sections.

Author Response

Manuscript Diagnostics - 1943755

Response to Reviewers

We would first like to thank both reviewers for their consideration of our revised manuscript “The Evaluation of 68Ga Radiolabeled Peptides for HER2 PET Imaging”. We have addressed the comments for each reviewer below.

Reviewer One

This manuscript reported the development and evaluation of 68Ga-radiolabeled peptides. This study was well performed and improved, but it still remains unclear whether these probes really work for visualizing HER2 high-expressing tumors in vivo. Thus, it should be reconsidered and revised in accordance with my opinion written below.

Response: Thank you for the positive feedback regarding the performance and improvement of our study.

  1. In the in vivo biodistribution study, the accumulation levels in BT474 tumor were higher than those in MDA-MB-231 tumor. Still, the levels in the other organs were also different depending on the mice types. Thus, it still remains hard to confirm if the different accumulation levels in tumor tissues depended on the HER2 expression levels of the tumors and not be affected by the pretreatment of beta-estradiol. Therefore, the biodistribution study should be performed with a mice model xenografting BT474 and MDA-MB-231 to avoid estimating the normal organs and the effect of the pretreatment.

            Response: Thank you for your feedback. Our in vivo tumor xenograft models involve implanting two well-characterized cell lines: MDA-MB-231, which is triple-negative and BT474, which is HER2 overexpressing. Previous studies have used estrogen-containing pellets to aid in the growth of the tumors indicating this is an established procedure when evaluating BT474 tumors. Our data shows that we have significantly higher binding to our HER2 overexpressing tumors compared to the triple-negative tumors. This supports that both peptides were able to target HER2 as that is the primary difference between the two models. There were no significant differences in each organ uptake between the different tumor xenograft groups except for the tumor, which also supports the conclusion that the expression of HER2 would be involved with the increase in radiotracer binding.

  1. Regarding PET imaging study, this manuscript showed only axial images, but MIP or coronal images would be more helpful to understand the distribution of those probes in the whole body.

            Response: We agree with the reviewer that other image slices would be helpful to understand the distribution of our probes. Supplemental figure 8 has the coronal, sagittal, and transverse slices for our PET images reference in our manuscript.

  1. According to the results of the PET imaging (figure 7), the distribution of [68Ga]P6 seems to be heterogeneous in the tumor tissues. Thus, it would be helpful to confirm the distribution of the probe was dependent on the HER2 expression by comparing autoradiography and HER2 immunohistochemistry with serial frozen sections.

            Response: We would like to thank the reviewer for their response. We have reported that we had modest uptake in our tumors and future studies would need to focus on the improvement of these peptides to increase stability and binding affinity. Image analysis indicated non-significant but higher SUVmean values in our HER2 positive tumors compared to our HER2 negative tumors. The non-significant difference is likely due to the modest binding and lower sensitivity of the PET scanner compared to the biodistribution data. With the use of established tumor models, supporting biodistribution data, and SUV analysis, our data shows that our peptides bind to HER2.

Reviewer 2 Report (Previous Reviewer 1)

All problems have been solved.

Author Response

Manuscript Diagnostics - 1943755

Response to Reviewers

We would first like to thank both reviewers for their consideration of our revised manuscript “The Evaluation of 68Ga Radiolabeled Peptides for HER2 PET Imaging”. We have addressed the comments for each reviewer below.

Reviewer Two

All problems have been solved.

            Response: We would like to thank the reviewer for their support and agreement that our suggestions for the manuscript have been address and that the manuscript is suitable for publication.  

This manuscript is a resubmission of an earlier submission. The following is a list of the peer review reports and author responses from that submission.

Round 1

Reviewer 1 Report

The authors modified two previously discovered HER2 targeting peptides with the chelator DOTA and a PEGlinker resulting in DOTA-PEG212GSGKCCYSL (P5) and DOTA-PEG2-DTFPYLGWWNPNEYRY (P6) and  they were labeled with 68Ga and was evaluated for HER2 binding using in vitro cell studies and in vivo tumor xenograft modelsThe present study is worthy, but several drawbacks should be mentioned.

1. There is only one figure showing the tumor. Although you declared there was difference of uptake in two Her2 heterogeneity tumors, I cannot see obvious uptake from your figure. Additionally, please confirm whether these two figures had the same threshold as I saw quite different background uptake.

2. The bio-distribution of the tracer was shown in table. However, as a potential clinical used tracer. Please provide the whole body image (sagittal, transverse an coronal) instead of only transverse one. If possible, a series and MIP image at different timepoint would be better.

3. The discussion is not sufficient. Please compare other Her2 tracer in more details.

4. The authors investigated both two tracers, and the results were similar. Please discuss which one do you think has more potential for clinical use?

5. Please provide the chemical structure of these two tracers.

Reviewer 2 Report

This manuscript reported the development and evaluation of 68Ga-radiolabeled peptides. This study was well performed, but it seems to be quite challenging to confirm the 68Ga-radiolabeled peptides really have a high and specific affinity to HER2 from the results of this manuscript. Thus, to be published on diagnostics, it should be redesigned and then performed to prove if 68Ga-radiolabeled peptides have a potential for PET imaging of HER2 expression.

1. In this manuscript, 68Ga-radiolabeled peptides were developed based on the peptides of "KCCYSL" and "DTFPYLGWWNPNEYRY" which were reported previously. Regarding the affinity to HER2, the first peptide was quite low compared to the second one (disassociation constant: 30,000 nM vs. 18.6  nM). I wonder why the author chose those peptides with quite different affinities. In addition, another 68Ga-radiolabeled peptide has already been reported (ex. Bioorg Chem. 2021;106:104474), and thus, it should be added to the discussion about the comparison with those probes previously reported.

2. In the in vitro stability study, the probes became stable in plasma by co-incubation with phosphoramidon, a protease inhibitor (figure 3, S4). The strategy was reasonable, but it remains uncertain if this strategy could work in the in vivo study or in the future, in human studies, because it is unclear how much the inhibitor is needed for protecting those probes in the mouse or human, and whether the inhibitor has toxicity.

3. In the in vitro study, the author seems to confirm the specific affinity of the 68Ga-radiolabeled peptides from the cellular uptake study (figure 4). On the other hand, the cellular uptake study with excess cold peptides showed no significant difference between treated and non-treated groups (figure S6). Considering that the different types of cells (BT474 and MDA-MB-231), it should be considered that the significantly different cellular uptake was dependent not on the HER2 expression levels but on the cell type. Therefore, this study should be reevaluated their specific activity in HER2 by such as measuring their disassociation constant against HER2 or performing cellular uptake studies with HER2 transfecting or knockdown cells.

4. In the in vivo biodistribution study, the accumulation levels in BT474 tumor were higher than those in MDA-MB-231 tumor, but the levels in the other organs were also different depending on the mice types. Thus, it remains quite hard to confirm if the different accumulation levels in tumor tissues depended on the HER2 expression levels of the tumors. Therefore, the biodistribution study should be performed with a mice model xenografting both BT474 and MDA-MB-231 to avoid the estimation of the normal organs.

5. In the in vivo PET imaging study, the SUV values of BT474 were a little bit higher (figure 7). However, as mentioned in comment 4, this study was performed with different mouse models, which could lead to the different biodistribution in normal tissues. Thus, the PET imaging study should be performed with a mice model xenografting both BT474 and MDA-MB-231. In addition, to confirm whether the 68Ga-labeled probes could accumulate in tumors dependent on the HER2 expression levels, a comparison study of autoradiography and  HER2 immunohistochemistry with serial frozen sections should be performed.

Round 2

Reviewer 1 Report

The authors revised the paper.

Reviewer 2 Report

The authors considered l and revised well in accordance with my comment. However, it has still remained unclear whether the 68Ga labeled probes ([68Ga]P5 and [68Ga]P6) developed in this study have the potential to HER2 targeting PET probes from the present data. Especially, it should be considered that the in vivo data is not enough for evaluating the specific affinity to HER2 high-expressing tissues/regions. Thus, if those probes are wanted to be defined as HER2 targeting PET probes, the experiment, especially in vivo study should be redesigned and performed.